# Predicting Out-of-Stock Using Machine Learning: An Application in a Retail Packaged Foods Manufacturing Company

**Juan Manuel Rozas Andaur** [1,*]**, Gonzalo A. Ruz** [1,2] **and Marcos Goycoolea** [3]

1   Facultad de Ingeniería y Ciencias, Universidad Adolfo Ibáñez, Diagonal Las Torres 2640, Peñalolén, Santiago 7941169, Chile; gonzalo.ruz@uai.cl

2   Center of Applied Ecology and Sustainability (CAPES), Santiago 8331150, Chile

3   Business School, Universidad Adolfo Ibáñez, Peñalolén, Santiago 7941169, Chile; marcos.goycoolea@uai.cl

*   Correspondence: jurozas@alumnos.uai.cl; Tel.: +56-9-56388994

**Abstract:** For decades, Out-of-Stock (OOS) events have been a problem for retailers and manufacturers. In grocery retailing, an OOS event is used to characterize the condition in which customers do not find a certain commodity while attempting to buy it. This paper focuses on addressing this problem from a manufacturer's perspective, conducting a case study in a retail packaged foods manufacturing company located in Latin America. We developed two machine learning based systems to detect OOS events automatically. The first is based on a single Random Forest classifier with balanced data, and the second is an ensemble of six different classification algorithms. We used transactional data from the manufacturer information system and physical audits. The novelty of this work is our use of new predictor variables of OOS events. The system was successfully implemented and tested in a retail packaged foods manufacturer company. By incorporating the new predictive variables in our Random Forest and Ensemble classifier, we were able to improve their system's predictive power. In particular, the Random Forest classifier presented the best performance in a real-world setting, achieving a detection precision of 72% and identifying 68% of the total OOS events. Finally, the incorporation of our new predictor variables allowed us to improve the performance of the Random Forest by 0.24 points in the F-measure.

**Keywords:** out of stock; machine learning; classification algorithms; imbalance data; supply chain management; decision support; retail industry application

## 1. Introduction

Technological advances and globalization are resulting in new challenges for retail operations [1,2]. An important challenge concerns ensuring on-shelf product availability, which significantly impacts customer loyalty and product demand [3,4]. Product availability is formally characterized in terms of Out-Of-Stock (OOS) events. An "OOS event occurs when, for some contiguous time, an item is not available for sale as intended. The OOS event begins when the final saleable unit of a stock-keeping unit (SKU) is removed from the shelf, and it ends when the presence of a saleable unit on the shelf is replenished" [5].

Despite efforts to reduce the number of OOS events, these are widely prevalent and result in important losses to the retail industry. In [6], it is estimated that 4% of total annual revenue worldwide in the retail sector is lost due to OOS events. In a more recent study [6], this loss is close to 984 billion dollars. OOS events result in a direct sale loss for retailers and manufacturers and operational and strategic costs. From an operational perspective, OOS events reduce the potential impact of promotions and distort true demand. From a strategic perspective, OOS events impact brand loyalty, promote competitors' brands and diminish the effectiveness of sales team resources [5].

OOS events affect retailers and manufacturers in different ways. Retailers typically manage a much larger number of SKUs and focus on key products that have a high share

in revenue or profit. They also count real-time transactional data for all SKUs and staff in point of sales that can be used to conduct physical audits [5]. Manufacturers must work with whatever data are provided by retailers and with limited sales staff if any.

A natural way for manufacturers to detect and predict OOS events without physical audits is to detect anomalies in the transactional data provided by retailers. It has recently been observed that such a process can be automated with a Machine Learning (ML) algorithm [7–9]. This approach was first pioneered by [8]. In [9], inventory control models and classification methods were studied. Different classification algorithms were compared in [7], and an ensemble learning approach was used to deal with the class imbalance problem. Other ways to detect and predict OOS events using transactional data have been reported in [10,11]

OOS continues to be a topic of research interest. Recently published studies have addressed the OOS problem from different approaches: image processing methods [12], autonomous robotic system to shelf monitoring [13], mobile robot, depth cameras and neural networks to determine the occupancy of a shelf [14], deep learning OOS detection [15], shelf monitoring using supervised learning [16], and a combination of semi-supervised learning and deep learning to monitor on-shelf availability (OSA) [17]. In [18], the authors studied the impact of automatic replenishment on product availability.

In this paper, we study the effectiveness of machine learning in predicting and detecting the OOS events of a manufacturer selling multiple products in a multiple-store retail operation. Specifically, we: (i) address the OOS problem from a manufacturer's perspective, using point-of-sale data shared by the retailer to manufacturer; (ii) identify important novel variables for predicting OOS events; (iii) measure and quantify the impact of the new predictor variables; (iv) compare the effectiveness of different ML algorithms; (v) compare the effectiveness of our calibrated approach with that of other approaches proposed in the literature; (vi) implement our approach in a retail packaged food products manufacturer, analyzing its performance in a real-world scenario. Although our methodology is very general, we focused our study on a nuts and dried fruits manufacturer that sells retail packaged products in an important grocery chain in Latin America. In this context, we found it possible to detect a significant amount of the OOS events (68%) with high precision (72%). In addition, we found that the novel variables proposed in this work were at the top of the information gain. Novel inventory and ordering predictive variables were relevant and contributed to the model's predictive performance. Furthermore, our model was successfully implemented at the manufacturer.

The methodology that we propose differs from previous approaches in the academic literature [7–11] in several ways. Previous approaches all use sales data to detect OOS events. This is natural, given that when sales fall below forecast, OOS events are a likely explanation. However, our study goes beyond this and considers other variables available in standard retail information systems such as inventory variables (on shelf, storage, transit, and in distribution centers), orders, delivery and receptions variables (dates, quantities) and others that could also be used to detect OOS events. The only other studies to use ML to classify such as we do are those of [7–9]. These studies adopt a retailer's point of view, whereas we adopt a manufacturer's point of view. Other studies, such as those of [10,11], obtain results comparable to ours, use different methodologies such as demand distribution estimations with consecutive zero-sales signals and a Hidden Markov Model.

Finally, we remark that our approach can easily be implemented with R using standard data provided by retailer's systems.

The paper is organized as follows. Previous studies related to the out-of-stock problem, detection mechanisms, and the classification algorithms used are reviewed in more detail in Section 2. In Section 3, we describe our methodology. We describe how we collected the data used in the study, selected the predictor variables, balanced the data, and ran our classification algorithms. The computational results of the study are presented in Section 4 and discussed in Section 5. Finally, Section 6 includes conclusions and future research directions.

## 2. Literature Review

The study of OOS events has been an active area of research for more than fifty years [2,19,20]. Papers in this field can broadly be classified into three groups: (i) Those that study the drivers of OOS events; (ii) those that study how consumers respond to OOS events; and (iii) those that study how to predict and detect OOS events.

### 2.1. The out of Stock Problem: Drivers and Consumer Response

For the past decades, the topic of OOS events has been a focus of study [19] and is part of the field of study in retail operations [2]. Given the importance of OOS events, different areas of research have addressed this problem. Research in Marketing and Consumer Behavior has studied the consumer response when faced with an OOS situation. Some of the proposed consequences are consumer negative reactions, store-switching, product-switching or purchase postponement [21–26]. Another area of research focuses on studying OOS drivers. In recent years, [20] presented a systematic review about drivers of retail on-shelf availability. The drivers can be categorized into two main groups, retail store practices and upstream problems in the retail supply chain. Some examples for the first group are inventory inaccuracy, shrinkage, and poor shelf replenishment processes. The second group is shrinkage due to product handling and transportation and forecast inaccuracy [4,5,27,28].

Related to retail store practices, [29] investigates the problems generated by inaccurate information in inventory systems. In this research, the authors demonstrated that inventory inaccuracy generates out-of-stock situations, which affect the replenishment system [30]. Indicate that a key task for retailers is product selection, and the shelf space allocated to each product and product positions architecture is commonly connected to shelf out-of-stock incidents. Their work proposed a strategy to optimally re-allocate shelf space to reduce the number of OOS events. In [31], the authors suggested that retail operations and in-store logistics are key areas to improve on-shelf availability but can only be done by modifying current processes. With advances in technology, RFID (Radio Frequency Identification) has been used in retail operations in the last decade, [32] studies how useful RFID is for automating shelf replenishment decisions in a retail store, monitoring the movement of products between the backroom and the shop floor.

From a supply-side characteristic, we can find two types of distributions: traditional (i.e., the manufacturer delivers products to the retail distribution center (DC) and this distributes to the stores) and direct store delivery (DSD) (i.e., the manufacturer delivers products directly to the retailer's individual stores). In [33], the authors integrated demand and supply-side issues to determine the consequences of repeated OOS conditions. They proposed a negative impact on the manufacturer and retailer when faced with repeated OOS conditions. Furthermore, they found that, from the supply-side, the retailer and the manufacturer could benefit from a DSD distribution strategy when facing repeated OOS events. However, DSD entails higher costs compared to the traditional distribution strategy. Another line of work has focused on studying the relationship between attributes of the SKU and stock-outs performance. [34] proposed that one of the causes that negatively affect stock-outs performance is the presence of fast-selling items. On the other hand, a positive result by reducing the number of out-of-stocks has been achieved with the automation of the ordering process. Product availability is also related to forecasting [5,35]. proposed the application of echelon inventory policies to reduce short-term order forecast error. [36] proposed using a hybrid artificial neural network to develop a sales forecasting model for fresh food. The model's predictions allow the identification of whether there are either insufficient or too many products in-store. [37] focused on the interrelationships between retail supply chain and marketing variables. They suggested that larger case packs decreased the number of store replenishments and, as a result, the store's chance of stockouts decreases. Another result of this study indicated that combining retail marketing and supply chain management decisions is crucial to the sustainability of customer-focused companies like retailers.

Considering the different factors related to the occurrence of OOS, [1] proposed to design and implement a participatory approach for consistently reducing OOS events. This methodology included varied parameters such as Demand Data, Replenishment Cycle, and Packaging System. The authors also commented that the multi-faceted nature of OOS events affects different companies in the supply chain, such as manufacturers, retailers, and stores.

### 2.2. Out of Stock: Detection and Prediction

Most research on the detection of OOS has been approached from the retailer's perspective. [38] presented the use of *zero balance walks*. In this strategy, the employees walk the store periodically to check for stockouts (physical audits). [39] implemented an RFID system in a retailer to monitor on-shelf availability. [7,11] used point of sale data (POS data) obtained directly from a retailer information system to detect the occurrence of OOS. Physical audits and RFID to detect OOS have in common their high overall cost, making them barely scalable to a large number of stores, categories, and products. Alternately, the detection of OOS events based on POS data (data-driven) present important advantages such as reducing labor intensity and human error of measurement [5], making this method scalable and more efficient in terms of overall cost. Some of the challenges of this method are access to the historical POS data, determining which variables are related to the OOS occurrence and developing mechanisms with more efficient detection performance. Usually, the POS data used are product sales (captured by check-out scanners) and inventory records. There is a lack of research in OOS detection from a manufacturer's perspective. In [10], they partnered with a product manufacturer and a retail service provider to use transactional data shared by the retailer to detect OOS events and correct them with physical audits.

Due to the OOS research generated in recent years, knowledge about their consequences and drivers has increased. This has led to the development of a third area of research, which addresses the detection and/or prediction of OOS occurrence, one of the most important challenges related to this problem [5]. In principle, [5] presented three methodologies for measuring OOS: manual audit method, POS sales estimation, and perpetual inventory aggregation. The first method is the traditional approach, where an auditor looks for "holes" generated by the products that are not visible on the shelf to the consumer. The second method uses point of sale data (POS data) to predict missed revenue due to OOS. Finally, the third method uses perceptual inventory data (PI), "PI systems track sales, and when sales = 0, the item is OOS" [5]. In recent times, new technologies and tools have been incorporated to measure and detect OOS, such as RFID, automated identification systems through image recognition, stochastic prediction models, and machine learning techniques.

As mentioned previously, [38] presented the *zero balance walk* method, where employees walk the shop floor regularly to look for stockouts. Some disadvantages of this method include that retailers often choose a small number of items and set a time for conducting these audits due to financial limitations and the lack of staff available to conduct physical audits. As a result, the expense of this approach is prohibitive for maintaining a continuous measurement, and therefore it is not scalable to a broad range of stores or products [5]. Currently, the retailer (internal audit) or the manufacturer (external audit) may start and lead the OOS measurement. The manufacturer can perform the audit either directly through their own work teams or via third-party, as retail service providers [10]. PI also has disadvantages since its accuracy is less than 50%, and it normally only detects store OOS rather than shelf OOS [5].

Reference [39] presented a pilot project that proposed an RFID infrastructure that stored data in real-time, allowing for OSA and OOS tracking, demonstrating the advantages of inventory management in a retail store. However, some authors have stated limitations of the use of this technology. [40] addressed that, owing to the physical limits of RFID, this approach has the potential to produce false negatives. To more precisely identify the OOS, each product must be identified and monitored. [41] stated that, for many applications,

item-level RFID implementations are also prohibitively costly, and they rarely have perfect insight into inventory locations. This is a major barrier to its implementation and scaling. RFID is still more expensive than conventional recognition technology, and its applications often come at higher costs [42]. It is for this reason that [43] wonders: "Should retail stores also RFID-tag 'cheap' items?". In the image recognition area, [44] presented some problems in detecting items from store shelves. For example, due to related variations in size, posture, perspective, and others, object appearance is highly variable.

Related to POS data prediction, store scanners (for example, those found at the checkout of stores and record sales) and inventory data from the retail information system are used in OOS detection systems. When tested by manual audits, this approach has 85–90% accuracy in identifying true positives (items that were correctly marked as OOS). However, the high rate of false negatives is a drawback (items that are OOS but are not detected as OOS). Other limitations of this approach include not handling low-turnover SKUs well and the need for precise POS data. Despite this, it is a viable solution to manual auditing, as it improves the efficiency of human resources, lowers the rate of estimation of human error, and is scalable [5].

The use of inventory data has some limitations, related particularly to its inaccuracy. [45] showed that eliminating inventory inaccuracy would lower out-of-stock levels. Probabilistic approximations and statistics have recently been proposed to detect OOS. [46] presented the use of Bayesian probability to correct "*phantom stockout*", using historical data from inventory records and sales, together with inventory inspection policies. The authors also considered the "*Bernoulli shrinkage process*" and a threshold for consecutive zero-sales periods.

Reference [10] used historical POS data to detect possible shelf OOS in a case study. This knowledge is used to send auditors to the stores to rectify any OOS issues. This paper further explains how a manufacturer should collaborate with a service provider to implement external physical audits to increase on-shelf availability. The demand is described using a negative binomial model, and OOS detection is conducted using consecutive zero-sales. Similarly, [10,47] proposed creating an optimal audit strategy, utilizing consecutive zero-sales measurements in POS data. [11] presented a proposal to predict the occurrence of OOS through the application of a hidden Markov model (HMM). In this model, an OOS condition is represented by one of the hidden states. The other states detect changes in the demand patterns. POS data is used to calibrate the model, and it is validated through shelf audits.

Reference [8] is one of the first works to use machine learning techniques and data obtained from the retailer to detect missing products on the shelf, proposing the development of a Decision Support System that automatically detects shelf-OOS. The data are divided into two categories: Data obtained from a retailer's internal information system (POS-data, Product assortment, Product categories, Product catalog, and Orders); and data obtained from physical audits. The OOS prediction is posed as a binary classification problem. With the data obtained from the information system, different attributes were calculated, which are part of the independent variables of the problem. The dependent or output variable is obtained from the physical audit, identified as EXISTS (product available on the shelf) and OOS (product not available on the shelf). [8] also presented significant challenges as they search for solutions that improved the trade-off between Support and Accuracy. Another challenge is to conduct more studies to determine which variables are more important in predicting OOS. Finally, this research opens a collaborative supply chain opportunity since both the retailer and the manufacturer can use this type of system. [9] examined two methods for developing a decision system that allows store managers to reduce the out-of-shelf rate. One of these methods utilizes inventory control models, and the other is a classification method. An important conclusion of this work was that the decision support tools should incorporate both methods, which could be key to improving decision support systems in retail. Concerning the research methodology presented in [7,8], they compared different classification algorithms to classify 'out-the-shelf' items in a real-life scenario. This

real-life data presented a difficulty in that OOS is a minority class, which implies a class imbalance problem for the classification algorithm. To overcome this problem and improve the efficiency of the classifiers, an ensemble learning approach is used. A limitation of the system is that the prediction system did not identify a large portion of the out-of-shelf items. Similar to what was exposed in previous works, it is necessary to understand the factors that affect product availability to develop tools with greater detection capacity.

## 3. Methodology

The methodology used in this work consists of five stages summarized in Figure 1. In what follows, a detailed description of each stage is given.

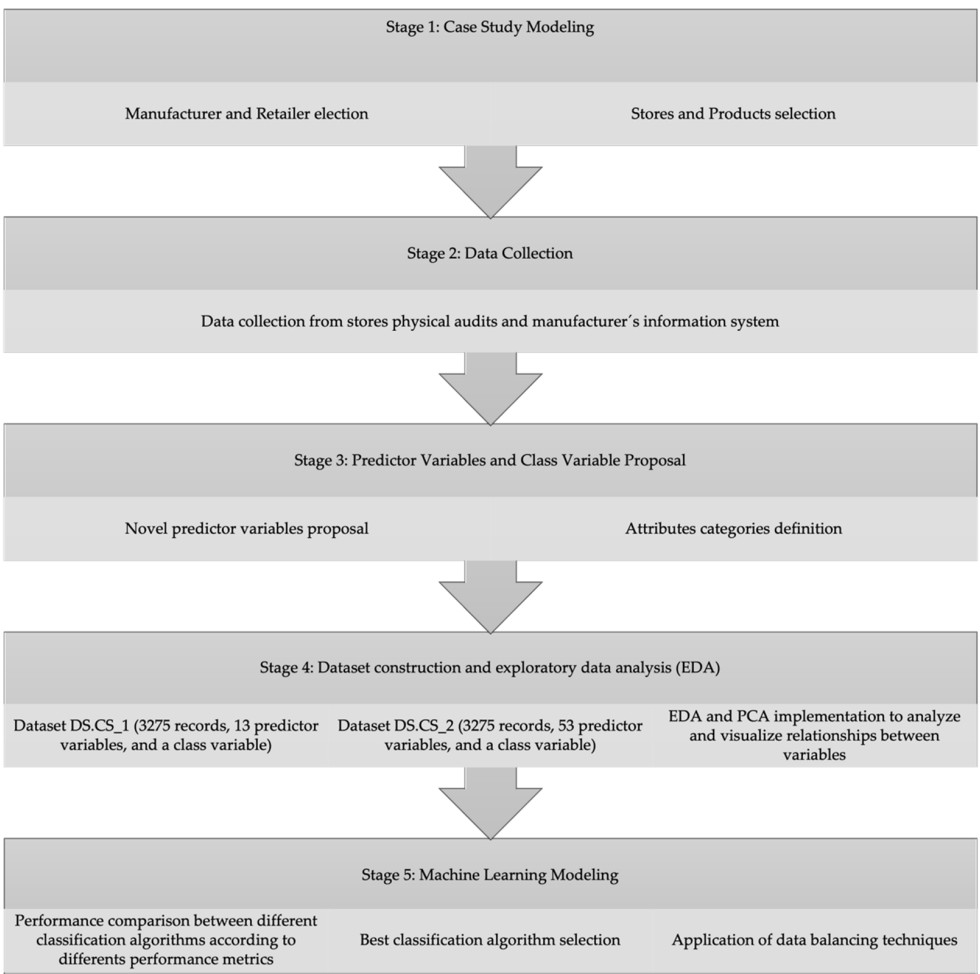

**Figure 1.** Methodological framework.

### 3.1. Case Study Modeling

The real-life problem that we address in this paper is the detection of out of stocks in the retail industry. This problem affects both the retailer and the manufacturer. As mentioned before, different proposals have been developed to detect or predict the occurrence of OOS [7,8,10,11]. This work proposes addressing this problem from the manufacturer's perspective, using machine learning algorithms for classification, incorporating new predictor variables computed from POS data obtained from the manufacturer's information system. The first stage is modeling the case study, converting the OOS real-life problem into a mathematical problem, which a classification algorithm could solve. The case study presented in this work is carried out in a manufacturing company that supplies packaged nuts and dried fruits products to a big box retailer located in Latin America. The OOS detection is carried out by using POS historical data. The input variables correspond to

sales, inventory, purchase orders, and purchase receptions. The output variable is binary: EXIST or OOS. The first is used if there is a product on the shelf and the second, in the opposite case. Once the data are obtained, an exploratory data analysis (EDA) is performed to analyze the behavior of the proposed variables and evaluate their discriminatory power. Then, different classification algorithms are tested, comparing their performance using the following metrics: *Accuracy*, *Sensitivity*, *Positive predicted value* (*Precision*), *Specificity*, and *Negative predicted value*. The metrics *Sensitivity* and *Positive predicted value* are the most important metrics. *Sensitivity* measures OOS occurrence, and *Positive predicted value* measures if the OOS occurrence is being correctly detected.

The number of stores studied was 16. They were classified according to their size in ten hypermarkets (big stores) and six supermarkets (small stores). The number of products studied was 14. These belonged to the fruits and vegetable category and to the packaged nuts and dried fruits subcategory. The product shelf-life classification was perishables. They were all active items at the store level (products enabled to be purchased), they were not promotional products (this avoids bias that the product has been replaced somewhere other than the shelf assigned), the products replenishment was through Computer Assisted Ordering—CAO (the purchase orders issued to the manufacturer are placed by an automatic computer system) and their distribution system was cross docking (the manufacturer delivered products to the retailer's DC, who immediately delivered them to the stores).

*3.2. Data Collection*

The data used in this work were historical point-of-sale data (POS data) and stores' physical audits data (Pa.D). The POS data were obtained from the manufacturer's information system, which the retailer information system shared. The period of the data collection was six mobile months. The POS data incorporated four different datasets, Sales data (SD) contained sales operations records, Inventory data (ID) contained inventory operations records, Purchase orders (PO) contained purchase operations records, and Purchase receptions (PR) contained receiving operations records. These records were aggregated on day level basis, per product and store. Each dataset also had categorical information, such as product and store description, product assortment (active items non-promotional), and delivery method (cross docking). The Pa.D dataset was collected daily through physical audits performed on the stores by the manufacturer's auditors. This dataset had the class variable. The audit period was five consecutive weeks and, Sundays were not considered. The manufacturer's auditors visited the stores early in the mornings. They recorded whether the products were on the shelf or not, using a data collection form similar to the one presented in [8] (Figure 2). This form contained all the active products to be audited for each store per day. If the researcher found a product on the shelf, EXISTS box was checked. On the contrary, if a product was not on the shelf, the OOS box was checked. The Pa.D dataset contained 3275 records (rows), with nine attributes (date, hour, store number, store description, product number, product description, category, subcategory and week number) and a class variable.

Time of vist ___________
Store number ___________
Store name ___________

| | | | Available on Shelf | |
|---|---|---|---|---|
| SKU | Description | Category | EXIST | OOS |
| 00001 | Product 1 | FF VV | X | |
| 00002 | Product 2 | FF VV | X | |
| 00003 | Product 3 | FF VV | X | |
| 00004 | Product 4 | FF VV | | X |

**Figure 2.** Shelf availability Physical Audit Sheet.

### 3.3. Predictor Variables and Class Variable Proposal

In this work, we proposed novel predictor variables to detect OOS through classification algorithms, and we also selected predictor variables used in previous research [7,8]. In the first step, 68 attributes were defined. As shown in Table 1, these attributes were classified into seven categories—Sales, Inventory, Product, Context [7], Ordering, Supply chain, and Description features (new categories proposed):

- Sales features describe the sales of each pair product-store for a time period.
- Inventory features present information on inventory records, In stock, and adjustments for each pair product-store for a time period.
- Product features describe the product's inherent characteristics.
- Context features present information regarding external variables that impact certain aspects of the product.
- Ordering features contain information regarding the purchase orders issued to the manufacturer and received by the retailer for each product-store for a time period.
- Supply chain features describe logistics and replenishments methods.
- Description features present categorical data for the identification of products, stores, and purchase orders. It also includes date data.

**Table 1.** Predictor variables definition, selection, and dataset details.

| Attribute Category | Attribute Description | DS.Initial Attribute Name | DS.CS_1 Predictor Variables Selection | DS.CS_2 Predictor Variables Selection | Predictor Variables Source | References |
|---|---|---|---|---|---|---|
| Sales features | Items sold for the specific day | SF 1 | YES | YES | Information system | [7,8] |
| | Average number of items sold in a period | SF 2 | YES | YES | Computed | [7,8] |
| | Standard deviation of items sold in a period | SF 3 | YES | YES | Computed | [7,8] |
| | Average items sold for a specific day of the week in a period | SF 4 | YES | YES | Computed | [7,8] |
| | Standard deviation of items sold for a specific day of the week in a period | SF 5 | YES | YES | Computed | [7,8] |
| | Average number of sales using only the days that a product made a sale in a period | SF 6 | YES | YES | Computed | [7,8] |
| | Standard deviation of sales using only the days that a product made a sale in a period | SF 7 | YES | YES | Computed | [7,8] |
| | The number of days that a product hasn't sold any single unit in a period | SF 8 | YES | YES | Computed | [7,8] |
| | The average number of days that the product is not selling a unit | SF 9 | YES | YES | Computed | [7,8] |
| | The standard deviation of days that the product is not selling a unit | SF 10 | YES | YES | Computed | [7,8] |
| | The number of days that a product hasn´t sold any single unit in a period/Total number of days in a period (percent) | SF 11 | YES | YES | Computed | New attribute |
| | The total number of days that a product hasn´t sold any single unit in a period | SF 12 | YES | YES | Computed | New attribute |
| | Median number of items sold in a period | SF 13 | YES | YES | Computed | New attribute |
| | Items sold for the specific day/Average number of items sold in a period (percent) | SF 14 | YES | YES | Computed | New attribute |

| Attribute Category | Attribute Description | DS.Initial Attribute Name | DS.CS_1 Predictor Variables Selection | DS.CS_2 Predictor Variables Selection | Predictor Variables Source | References |
|---|---|---|---|---|---|---|
| Inventory features | Inventory level for the specific day | IF 1 | NO | YES | Information system | New attribute |
| | Inventory level for the specific day before | IF 2 | NO | YES | Computed | New attribute |
| | Average inventory level in a period | IF 3 | NO | YES | Computed | New attribute |
| | Standard deviation inventory level in a period | IF 4 | NO | YES | Computed | New attribute |
| | Average inventory level for a specific day of the week in a period | IF 5 | NO | YES | Computed | New attribute |
| | Standard deviation inventory level for a specific day of the week in a period | IF 6 | NO | YES | Computed | New attribute |
| | Average inventory level using only the days that a product has positive stock in a period | IF 7 | NO | YES | Computed | New attribute |
| | Standard deviation inventory level using only the days that a product has positive stock in a period | IF 8 | NO | YES | Computed | New attribute |
| | In stock for the specific day | IF 9 | NO | YES | Information system | New attribute |
| | Stock adjustment for the specific day | IF 10 | NO | YES | Information system | New attribute |
| | Average Stock adjustment in a period | IF 11 | NO | YES | Computed | New attribute |
| | Standard deviation Stock adjustment in a period | IF 12 | NO | YES | Computed | New attribute |
| | Average Stock adjustment for a specific day of the week in a period | IF 13 | NO | YES | Computed | New attribute |
| | Standard deviation Stock adjustment for a specific day of the week in a period | IF 14 | NO | YES | Computed | New attribute |
| | The number of days that a product has been adjusted in a period | IF 15 | NO | YES | Computed | New attribute |
| | The number of days that a product has been adjusted in a period/Total number of days in a period (percent) | IF 16 | NO | YES | Computed | New attribute |
| | The number of days that a product has been positive adjusted in a period | IF 17 | NO | YES | Computed | New attribute |
| | The number of days that a product has been positive adjusted in a period/Total number of days in a period (percent) | IF 18 | NO | YES | Computed | New attribute |
| | The number of days that a product has been negative adjusted in a period | IF 19 | NO | YES | Computed | New attribute |
| | The number of days that a product has been negative adjusted in a period/Total number of days in a period (percent) | IF 20 | NO | YES | Computed | New attribute |

**Table 1.** *Cont.*

| Attribute Category | Attribute Description | DS.Initial Attribute Name | DS.CS_1 Predictor Variables Selection | DS.CS_2 Predictor Variables Selection | Predictor Variables Source | References |
|---|---|---|---|---|---|---|
| Product features | The product is a regular line product or a promotional item | PF 1 | NO | NO | Information system | [7,8] |
| | The product is new or regular | PF 2 | NO | NO | Information system | [7,8] |
| | Product category | PF 3 | NO | NO | Information system | [7,8] |
| | Product subcategory | PF 4 | NO | NO | Information system | [7,8] |
| | Product Shelf Life | PF 5 | NO | NO | Information system | New attribute |
| | Product Status | PF 6 | NO | NO | Information system | New attribute |
| Context features | The size of the store | CF 1 | YES | YES | Information system | [7,8] |
| | The day of the week | CF 2 | YES | YES | Information system | [7,8] |
| | Moving index | CF 3 | YES | YES | Computed | [7,8] |
| Supply Chain features | The logistics method for the specific product in the specific store (Product distribution) | SC 1 | NO | NO | Information system | New attribute |
| | The replenishment method for the specific product in the specific store (Product replenishment) | SC 2 | NO | NO | Information system | [7,8] |
| Ordering features | PO emission: number of items purchased on a specific day in the specific store | OF 1 | NO | YES | Information system | New attribute |
| | Average PO emission in a period | OF 2 | NO | YES | Computed | New attribute |
| | Standard deviation PO emission in a period | OF 3 | NO | YES | Computed | New attribute |
| | Average PO emission for a specific day of the week in a period | OF 4 | NO | YES | Computed | New attribute |
| | Standard deviation PO emission for a specific day of the week in a period | OF 5 | NO | YES | Computed | New attribute |
| | PO reception qty (deliver): number of items to deliver on a specific day to the DC for a specific store | OF 6 | NO | YES | Information system | New attribute |
| | PO reception qty (received): number of items received on a specific day to the DC for a specific store | OF 7 | NO | YES | Information system | New attribute |
| | Fill rate percent: number of item received/number of items purchased on a specific day in the specific store | OF 8 | NO | YES | Computed | New attribute |
| | Fill rate percent in a period | OF 9 | NO | YES | Computed | New attribute |
| | Average PO emission last 14 days | OF 10 | NO | YES | Computed | New attribute |
| | Standard deviation PO last 14 days | OF 11 | NO | YES | Computed | New attribute |
| | Fill rate percent last 14 days | OF 12 | NO | YES | Computed | New attribute |
| | The number of days that a product has been ordered in a period | OF 13 | NO | YES | Computed | New attribute |
| | The number of days that a product has been ordered in a period/Total number of days in a period (percent) | OF 14 | NO | YES | Computed | New attribute |

**Table 1.** *Cont.*

| Attribute Category | Attribute Description | DS.Initial | DS.CS_1 | DS.CS_2 | | |
| | | Attribute Name | Predictor Variables Selection | Predictor Variables Selection | Predictor Variables Source | References |
|---|---|---|---|---|---|---|
| | PO reception qty (deliver): number of items to deliver on a period to the DC for a specific store | OF 15 | NO | YES | Computed | New attribute |
| | PO reception qty (received): number of items received on a period to the DC for a specific store | OF 16 | NO | YES | Computed | New attribute |
| Description features | Product number | DI 1 | NO | NO | Information system | |
| | Product description | DI 2 | NO | NO | Information system | |
| | Store number | DI 3 | NO | NO | Information system | |
| | Store description | DI 4 | NO | NO | Information system | |
| | Date | DI 5 | NO | NO | Information system | |
| | PO emission number | DI 6 | | | | |
| | PO reception number | DI 7 | NO | NO | Information system | |
| Total number of attributes per datasets | | 68 | 13 | 53 | | |

From these attributes, in a second step, the predictor variables were selected. Previous works [7,8], show that the most important predictor variables correspond to sales features. They chose a few inventory features, but the inventory level was estimated, and adjustment operations were not considered. Ordering features were not considered either. From [7], we selected thirteen predictor variables, ten sales features variables, and three context features variables (see Table 1). The novel predictor variables proposed in this work were 40 and corresponded to sales, inventory, context, and ordering features. The new sales features variables were SF 11, SF 12, SF 13, and SF 14. The new inventory features variables were twenty, from IF 1 to IF 20. There were two new context features variables, CF 4 and CF 5. Finally, we propose sixteen new ordering features variables, from OF 1 to OF 16 (see Table 1). The importance of these new predictor variables is that they should improve the detection performance of the classification algorithms. As mentioned before, [7,8] showed that sales features were the main predictor variables to detect OOS. The occurrence of OOS generates a direct impact on product sales. If the product is not on the shelf, it could not be purchased. This causes distortions in the sales patterns that could be used to detect OOS. To strengthen this feature category, the new variables SF 11 and SF 12 were proposed. These were related to *zero sales* records (days with 0 units sold), SF 13 was the median sales value, and SF 14 presents a relationship between the day sales and the average sales in a specific period, for each product-store pair in a period of time. This could contribute to making better predictions in cases with outliers. Furthermore, we proposed new inventory features variables. The inventory record inaccuracy (IRI) is an actual problem in the retail industry, defined as a "*discrepancy between real inventory holdings and inventory records*", which commonly causes phantom stockout (undetected inventory shrinkage) and affects the inventory replenishment negatively [10,29,46]. When a phantom stockout occurs, no product could be replenished on the shelf, causing OOS. This phenomenon requires a human inspection to be corrected [46]. Therefore, we proposed that deviations in inventory records patterns could contribute to the OOS detection. These new variables that capture these behaviors were IF 1 to IF 9.

Consequently, since IRI is a real and permanent problem, there were also corrections to these deviations made by the retailer staff that was recorded in the information system

(adjustments). Hence, we also proposed variables that capture adjustments operations as new predictor variables (IF 10 to IF 20) because they directly reflect IRI problems. Finally, we proposed new predictor variables related to purchasing orders and purchase receptions (OF 1 to OF 17). One of the effects produced by IRI was inventory replenishment problems. These could be detected through deviations in the periodicity and quantity of the purchase orders, especially in COA systems. In our work, we posed a broader scope to address the OOS problem, focusing on the fulfillment of the manufacturer's deliveries. For this purpose, we incorporate product delivery and fill-rate variables, which is an indicator that measures the quantity delivered from the manufacturer to the retailer regarding what was requested. A low percentage of fill-rate and the non-emission of purchase orders could cause OOS because it affects the prompt replenishment inventory. In the third step, the class variable was defined based on [7]. We considered two classes—EXISTS (product is on the shelf) and OOS (product is not on the shelf). Therefore, our case study corresponded to a binary classification problem.

### 3.4. Dataset Construction and EDA

Therefore, the first data processing stage was to filter the dataset SD, ID, PO, and PR to select only products and stores under study. The datasets obtained did not present missing or duplicate values. Some values of *items sold for the specific day* or *inventory level for the specific day* presented errors in their value format, which were identified and corrected. The Pa.D did not show missing or duplicate values. The second stage was to build the DS.initial dataset. This was achieved by consolidating the five datasets SD, ID, PO, PR, and Pa.D, by joining three keys: date, store number, and product number. The new dataset had 68 attributes belonging to the seven categories presented in the previous section, a class variable, and 3275 records. The third, final stage was constructing two new datasets (DS.CS_1 and DS.CS_2) from DS.initial (see Figure 3).

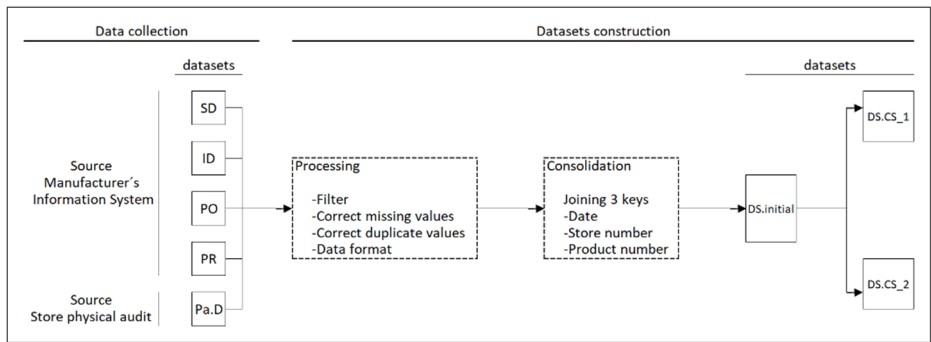

**Figure 3.** Data collection and datasets definition.

- DS.CS_1 contains only some of the predictor variables presented in [7]. The predictor variables selected were 13 and correspond to the main ones in terms of information gain. The DS.CS_1 dimension was 3275 records, 13 predictor variables, and a class variable.
- DS.CS_2 was built with 13 predictor variables selected from [7] and 40 new predictor variables proposed in this work. The DS.CS_2 contained 3275 records, 53 predictor variables, and a class variable. The unimportant variables, such as description features, were not considered in these three datasets.

Once the datasets were obtained, an EDA was performed to explore the relationship between the selected predictor variables and the class variable. We used a *Principal Component Analysis* (PCA) which is a statistical procedure for dimensionality reduction. This was implemented in R, using the packages *stats* [48] for PCA analysis and *factoextra* [49] for visualization.

*3.5. Machine Learning Modeling*

For the development of the machine learning based system, the performance of different classifiers was analyzed. The classification algorithms chosen were radial Support Vector Machines (rSVM) [50], Naive Bayes (NB) [50], Decision Tree (DT) [51], Regressions Logistics (RL) [48], Random Forest (RF) [52], neural networks (NNET) [53] and an Ensemble (ENS-Stack) of the previous six classification algorithms [54]. The technique that we applied to perform the ensemble was stacking [55]. We chose this technique because the models tested were different and were evaluated in the same dataset. To combine the predictions of the different models, we tested two single models, a linear model (linear regression) and a non-linear model (Random Forest). The linear regression model had a better performance, selecting this approach to combine the six classification models. All of these algorithms were implemented in R [48] using their respective packages. Some of the hyperparameters of the RF, DT, rSVM and NNET were optimized. NB and RL did not have hyperparameters that could be optimized. The rest of the parameters for each algorithm were set in *default* mode [48,50–54]. To optimize the hyperparameters, we use the training sets obtained from DS.CS_1 and DS.CS_2, and the Caret package [54] in R. For the tuning process we used a Cross-Validation method, with 5 folds. To select the best hyperparameters we use the *F-measure* [56]. The optimized hyperparameters were as follows:

- RF hyperparameters for DS.CS_1 was number of trees = 900 and number of variables to possibly split at in each node = 6. The hyperparameters for DS.CS_2 was number of trees = 600 and number of variables to possibly split at in each node = 6. The range of values studied for number of trees was 1 to 1000, and for the number of variables to test for a possible split in each node was 2 to 7.
- DT hyperparameter for DS.CS_1 was the maximum depth of any node of the final tree = 3. The hyperparameter for DS.CS_2 was the maximum depth of any node of the final tree = 4. The range of values studied for the maximum depth of any node of the final tree was 1 to 30.
- rSVM hyperparameter for DS.CS_1 was cost of constraints violation = 1 and sigma = 0.4. The hyperparameter for DS.CS_2 was cost of constraints violation = 1 and sigma = 0.3. The range of values studied for cost of constraints violation was 1 to 10, and for sigma was 0 to 1, in sequences of 0.1.
- NNET hyperparameter for DS.CS_1 was size = 20 (number of hidden units) and decay = 0.50. The hyperparameter for DS.CS_2 was size = 15 (number of hidden units) and decay = 0.53. The range of values studied for size was 1 to 25, and for decay was 0 to 1, in sequences of 0.01.

The performance comparison of the selected classification algorithms was performed for the two datasets built in this work: DS.CS_1 and DS.CS_2. The data partition approach used for both datasets was "holdout", where the training set corresponded to 70% and the test set to 30%. To compare the performance of the classification algorithms, the models were run 100 times randomly, and the results presented were the mean and standard deviation for each one of the performance metrics: *Accuracy*, *Sensitivity (Recall)*, *Positive predicted value (Precision)*, *Specificity*, *Negative predicted value* and *F-measure*. To determine the best classification algorithm, we used the trade-off between *Sensitivity (Recall)* and *Positive predicted value (Precision)*, using the *F-measure* metric.

After determining which was the best classification algorithm, the importance of the predictor variables was studied using permutation analysis.

We applied two balancing techniques to face the data imbalance data problem: over-sampling minority examples and under-sampling majority examples. These algorithms were obtained from the *ROSE* package [57]. The balanced data was tested in the Random Forest algorithm, following the same guidelines applied for the performance comparison of the classification algorithms.

*3.6. Performance Metrics*

The performance metrics were calculated from the confusion matrix [54]. The confusion matrix used is presented in Table 2. The positive class chosen was OOS because our goal is to predict the occurrence of OOS (minority class).

**Table 2.** Confusion Matrix.

| | | Actual | |
|---|---|---|---|
| | | OOS Shelf | EXIST |
| **Prediction** | OOS Shelf | True Positive (TP) | False Negative (FN) |
| | EXIST | False Positive (FP) | True Negative (TN) |

The performance metrics used in this study are:

- *Accuracy* was used to measure the overall prediction performance of the classification algorithms.

$$Accuracy = (TP + TN)/(TP + FP + FN + TN). \tag{1}$$

- *Sensitivity (Recall)** was used to determine the algorithm ability to detect correctly OOS products out of all the existing OOS in the store. *A high value of Sensitivity shows that the algorithm can detect a high number of OOS.*

$$Sensitivity = TP/(TP + FN). \tag{2}$$

- *Positive predicted value (Precision)** describes the number of times that the algorithm has correctly identified an OOS event as such. *A high value of Positive predicted value shows that the algorithm has a high detection power, presenting a low number of FP, allowing the system to be more efficient in the use of resources to correct OOS (controlling Type I error).*

$$Positive\ predicted\ value = TP/(TP + FP). \tag{3}$$

- *Specificity* was used to determine the algorithm ability to detect correctly EXIST products out of all the existing EXIST in the store. *A high value of Specificity shows that the algorithm has a low number of FP, allowing the system to be more efficient in the use of resources to correct OOS.*

$$Specificity = TN/(FP + TN). \tag{4}$$

- *Negative predicted value* describes the number of times that the algorithm has correctly identified an OOS event as such. *A high value of Negative predicted value shows that the algorithm is presenting a low number of FN, allowing us the system to be more efficient in the use of resources to correct OOS (controlling Type II error).*

$$Negative\ predicted\ value = TN/(TN + FN). \tag{5}$$

*Sensitivity** and *Positive predicted value** were the most important metrics used in this case study. An objective of this work and an important challenge in the retail industry is detecting the occurrence of OOS. *Sensitivity* is the metric that allowed us to determine this. The second objective is the implementation and use of this detection system in the real world. For this, it is important to control if OOS are correctly detected because this would trigger corrective actions by the manufacturer, which have an operational cost. To control this, the *Positive predicted value* was used. Therefore, the trade-off between these

two metrics is the most important measure for the application of our model in a real-world. To measure this trade-off, we used the performance metric *F-measure* [56],

$$F - measure = (2 \times Precision \times Recall) / (Precision + Recall). \tag{6}$$

## 4. Results

### 4.1. Case Study Modeling

In Figure 4, we present our case study modeling proposal. We adapted the supermarket layout shown in [2] to increase our case study scope. Our model begins with the manufacturer elaborating the products and dispatching them to the retailer's DC. The type of logistics addressed is Cross Docking. Therefore, there is no storage of the products in the DC since once received; they are delivered to the supermarket stores. The store's inventory registration system is updated to recognize the inventory entry when the products are received. Then, the products could be transferred to the backroom area or directly replenished on the shelf (shop floor). There may be a product flow from the shelf to the backroom. It occurs when the product cannot be replenished due to a lack of shelf space. The consumers who walk on the shop floor can access the products on the shelf, choosing which ones to buy. Finally, the consumers pass the products through the checkout, where their barcode is scanned. The total price is calculated. The purchase is finalized once they pay for their products (updating the inventory to recognize the inventory output). These retail operation activities are registered in the retailer's information system. The store operations (input variables) registered in the information system incorporated in our model are *sales operations*, which represent when a customer purchased a product at the store, and the product barcode is scanned at check out. *Inventory operations* include operations that generate inventory movements, such as product purchase (inventory decrease), product receipt (inventory increase), product return (inventory increases), and inventory adjustments. They also include inventory adjustments, which could be positive (inventory increase) or negative (inventory decrease). The first is executed when, for example, the store staff found products that were not inventoried (registered in the inventory system). The second is executed when there is a loss of inventory (shrinkage), for example, caused by theft or damaged products. *Purchase operations* are when the CAO automatically issues purchase orders to the manufacturer. Finally, *Receiving Operations* record when the manufacturer has delivered their products at DC, and these have also been received at the stores. The shelf status (output variable) collected through physical audits represents whether or not a product is on the shelf.

### 4.2. Data Set Construction and Exploratory Data Analysis (EDA)

Through the physical store audit and the construction of the Pa.D dataset, it was possible to determine the percentage of occurrence of OOS in five consecutive weeks. The result was a 10% occurrence of OOS (see Table 3). The main relevance of this result was that it is in accordance with the results presented in [58], where the occurrence of OOS is between 5% and 10%, with an average of 8.3%. Our result was also consistent with the information reported in [3], where OOS occurrence is between 10% and 15%. Another important observation regarding this result was the significant data imbalance, where the class variable (EXIST or OOS) was only 10% of the records obtained. The imbalance in our dataset was 4.6 points higher than the dataset presented in [7]. This characteristic in the distribution of the data obtained is not uncommon when working with real-world data. In [59], they present examples of class imbalance problems in real-world applications like credit card fraud detection, breast cancer diagnosis, and market segmentation. To analyze the performance of the classification algorithms, we decided to build two datasets. The dataset DS.CS_1 contains only the most relevant predictor variables proposed in [7] and the dataset DS.CS_2 included the DS.CS_1 predictor variables and the new predictor variables proposed in this work (Table 1). This approach was used to determine whether the most relevant predictor variables presented in [7], applied to a new case study dataset, could

obtain similar results. Furthermore, our principal goal is to determine if the inclusion of our new predictor variables allowed us to improve the performance of the classification algorithms.

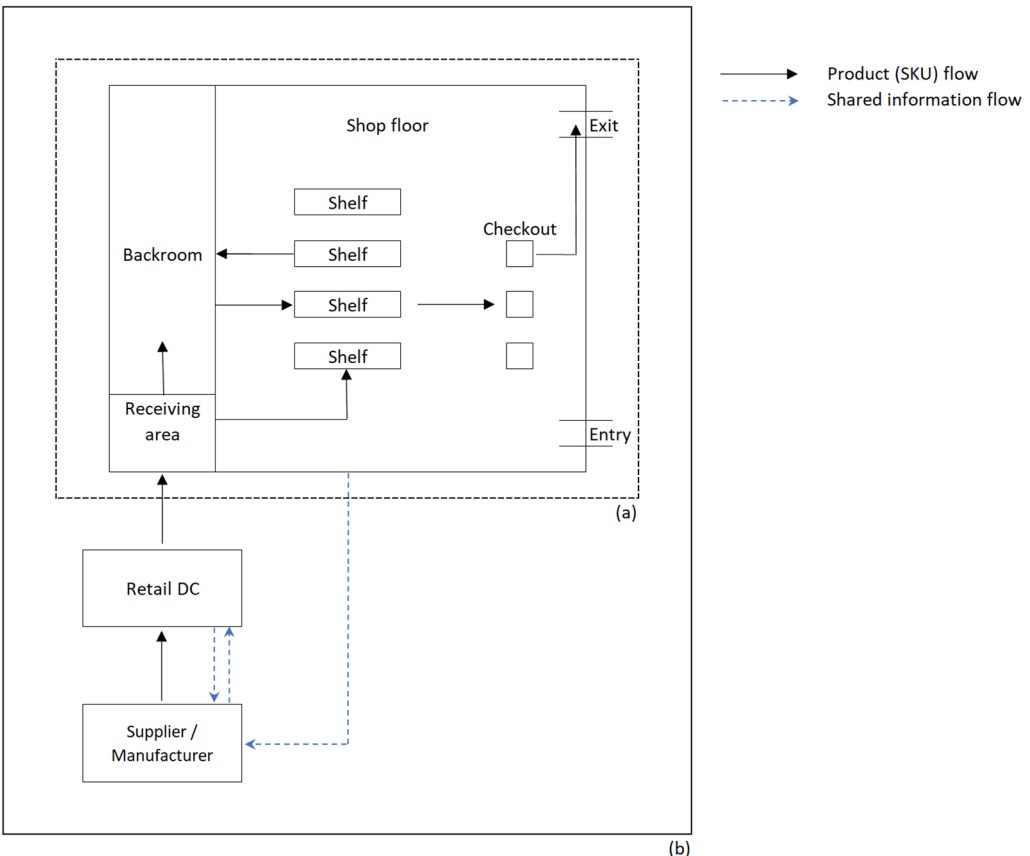

**Figure 4.** Case Study Modeling Proposal. (**a**) Simplified layout of a retail store [2]; (**b**) Complete model scope addressed in this work, starting from the Manufacturer and ending at the Supermarket store.

**Table 3.** Physical store Audit.

| Dataset | Description | Observed Class | | Total | Rate | Category | Product Characteristics |
| | | EXIST | OOS | EXIST + OOS | OOS | | |
|---|---|---|---|---|---|---|---|
| Pa.D | Store physical audit dataset | 2948 | 327 | 3275 | 10.0% | Fresh fruits and vegetables | Non-perishable food (Nuts and Dried fruit) |

Once the datasets are obtained, we executed an EDA on DS.CS_2. Figure 5 shows the relationship between some of the proposed predictor variables. In Figure 5a, a positive linear correlation is observed between the "Average inventory level in a period" and "Average number of items sold in a period" variables. This correlation could suggest that greater inventory availability helps to improve product sales because the store could better respond to consumer demand. In Figure 5b, a stronger positive linear correlation is observed between "PO reception qty (deliver): number of items to deliver in a period to DC for a specific store" and "Average number of items sold in a period" variables. This result could suggest that larger quantities of products delivered to retailers' DC contribute to better product sales. However, even when at higher predictor variables values, we observed that the data are classified in EXIST class (red points), at lower values, we observed that these pair of variables relationships were not sufficient to detect the occurrence of OOS

events, as shown by the classes overlapping (red and green points). The positive correlation observed between the new proposed variables, inventory and purchase order, and the average sales suggest that incorporating these variables could contribute to the detection of OOS events.

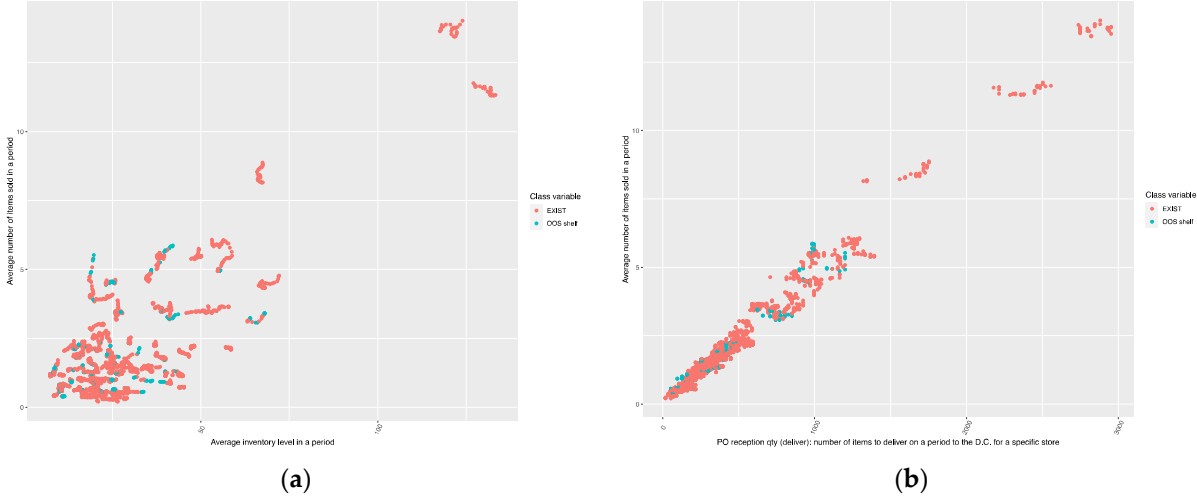

(**a**)                                              (**b**)

**Figure 5.** Exploratory Data Analysis (EDA) on DS.CS_2 Dataset. Relationship between predictor variables and class variable: (**a**) Average inventory level in a period vs. Average number of items sold in a period; (**b**) PO reception qty (deliver): number of items to deliver in a period to D.C. for a specific store.

We also performed a principal component analysis (PCA) to reduce the dataset dimensionality, analyzing whether linear combinations of variables allow us to describe the two classes present. As shown in Figure 6, principal components 1 and 2 capture 67.2% of the dataset variance. Notwithstanding, it is not observed that these components allow us to group the data according to the OOS and EXIST classes. This result suggests that the observed real-world data are not a linear combination, showing the problem complexity. Due to this complexity, it is necessary to use machine learning classification algorithms to detect the occurrence of OOS.

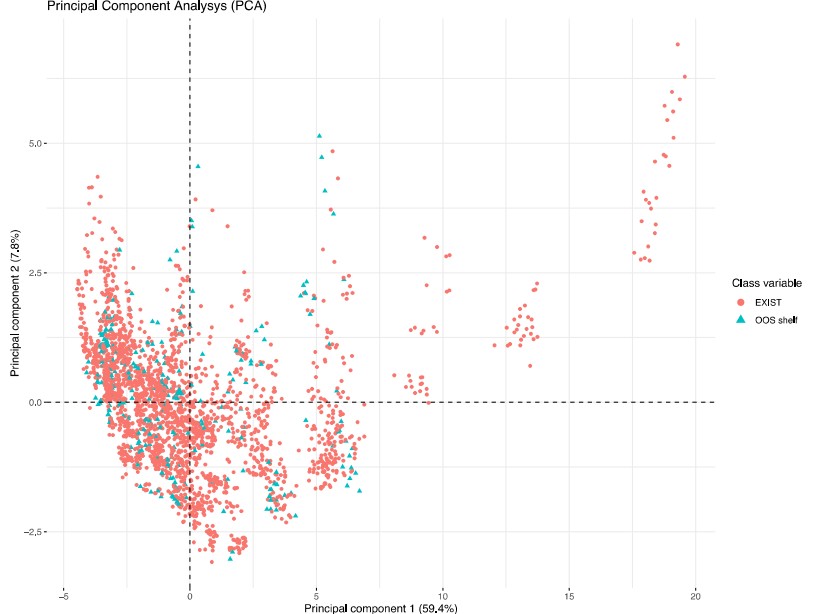

**Figure 6.** Principal Component Analysis (PCA) on DS.CS_2 Dataset.

### 4.3. Classification Algorithms Evaluation

Once the datasets were defined, we evaluated and compared the different classification algorithms. In previous works [7,8], efforts have already been made in this matter. In this work, we selected and compared seven classification algorithms: Random Forest, Logistic Regression, Decision Tree, Naive Bayes, Support Vector Machines, Neural Networks and an Ensemble of the previous six classification algorithms. The results obtained are presented in Table 4. One hundred random iterations were carried out, presenting the mean and standard deviation for each one of the performance metrics.

**Table 4.** Classification Algorithms Evaluation.

| | | DS.CS_1 | | | | | | DS.CS_2 | | | | | |
| --- | --- | --- | --- | --- | --- | --- | --- | --- | --- | --- | --- | --- | --- |
| | | Accuracy | Sensitivity | Pos. Pred. value | Specificity | Neg. Pred. value | F-measure | Accuracy | Sensitivity | Pos. Pred. value | Specificity | Neg. Pred. value | F-measure |
| Random Forest | mean | 0.91371 | 0.24164 | 0.73593 | 0.98991 | 0.92012 | 0.36381 | 0.93483 | 0.46179 | 0.85182 | 0.99054 | 0.93989 | 0.59890 |
| | sd | 0.00845 | 0.03856 | 0.08690 | 0.00424 | 0.00872 | 0.05341 | 0.00909 | 0.05364 | 0.05610 | 0.00385 | 0.00962 | 0.05484 |
| Logistics Regression | mean | 0.89786 | 0.00998 | 0.33848 | 0.99834 | 0.89910 | 0.01939 | 0.89378 | 0.11894 | 0.48455 | 0.98479 | 0.90493 | 0.19100 |
| | sd | 0.00811 | 0.01016 | 0.33726 | 0.00142 | 0.00817 | 0.01972 | 0.01066 | 0.03310 | 0.11306 | 0.00553 | 0.01061 | 0.05121 |
| Decision Tree | mean | 0.89719 | 0.10076 | 0.44402 | 0.98739 | 0.90657 | 0.16425 | 0.89838 | 0.27292 | 0.53902 | 0.97196 | 0.91924 | 0.36236 |
| | sd | 0.00858 | 0.05844 | 0.16184 | 0.00737 | 0.00968 | 0.08587 | 0.01099 | 0.07337 | 0.09683 | 0.00997 | 0.01194 | 0.08348 |
| Naïve Bayes | mean | 0.84537 | 0.19484 | 0.22353 | 0.91909 | 0.90979 | 0.20820 | 0.49551 | 0.70547 | 0.13648 | 0.47089 | 0.93096 | 0.22871 |
| | sd | 0.01885 | 0.03041 | 0.05639 | 0.02300 | 0.00782 | 0.03951 | 0.05285 | 0.04651 | 0.01934 | 0.06058 | 0.01383 | 0.02732 |
| radial Support Vector Machines | mean | 0.89768 | 0.02031 | 0.46870 | 0.99701 | 0.89990 | 0.03893 | 0.89718 | 0.03019 | 0.77800 | 0.99900 | 0.89767 | 0.05812 |
| | sd | 0.00829 | 0.01287 | 0.28384 | 0.00244 | 0.00849 | 0.02461 | 0.01046 | 0.01397 | 0.26620 | 0.00121 | 0.01081 | 0.02655 |
| Neural Networks | mean | 0.89943 | 0.07371 | 0.55110 | 0.99293 | 0.90447 | 0.13003 | 0.89631 | 0.13215 | 0.53547 | 0.98603 | 0.90640 | 0.21199 |
| | sd | 0.00853 | 0.02628 | 0.13326 | 0.00311 | 0.00865 | 0.04391 | 0.01102 | 0.06019 | 0.17137 | 0.00902 | 0.01110 | 0.08909 |
| Ensemble (stacking-linear regression) | mean | 0.91667 | 0.31467 | 0.70757 | 0.98497 | 0.92689 | 0.43561 | 0.93967 | 0.56735 | 0.80480 | 0.98358 | 0.95072 | 0.66553 |
| | sd | 0.00844 | 0.04419 | 0.07698 | 0.00534 | 0.00874 | 0.05615 | 0.00880 | 0.05451 | 0.05792 | 0.00590 | 0.00901 | 0.05616 |

According to the results obtained (Table 4), the algorithm with the best performance was the Ensemble (ENS-Stack). Its *Accuracy* metric was the highest, and the tradeoff between the most important metrics, *Sensitivity,* and *Positive predicted value (F-measure)*, was the best. These results were consistent for both datasets. For most classification algorithms, *Accuracy* presented similar high mean values with a low standard deviation. This is explained due to the data imbalance, where the majority variable EXIST contributes to obtaining better overall performance. The previous result was only not observed in the Naive Bayes algorithm applied in DS.CS_2, its *Accuracy* was (0.49551). Considering the objective of detecting a high number of OOS products with high exactness, the tradeoff between *Sensitivity* and *Positive predicted value* was fundamental. In the DS.CS_1 dataset, the Ensemble presented the highest value for *Sensitivity* (0.31467), and the second highest value for *Positive predicted value* (0.70757), consequently its *F-measure* was the highest (0.43561). In DS.CS_2, the highest value of *Sensitivity* (0.70547) was obtained in Naive Bayes, but this classification algorithm presented the lowest performance in *Positive predicted value* (0.13648). The Ensemble showed the second highest value for *Sensitivity* (0.56735) and the best tradeoff between and *Sensitivity Positive predicted value*, reaching a *F-measure* value of 0.66553. In general, for Logistic Regression, Naive Bayes, Support Vector Machines and Neural Networks, it was observed that they presented a poor tradeoff between *Sensitivity* and *Positive predicted value* (low *F-measure* value). These results showed how challenging it was to find a good balance between both performance metrics to develop a machine learning based system that can be applied in a real world setting.

Considering a trade-off performance between *Sensitivity* and *Positive predicted value*, the best single classification algorithm was Random Forest, however the Ensemble presented the best performance with approximately 0.06 points (*F-measure*) above Random Forest. This result is consistent with [7]. When we compared the Ensemble performance between the two studied datasets, DS.CS_2 showed a better *F-measure,* approximately 53% higher than DS.CS_1. This improvement observed in the Ensemble trade-off between *Sensitivity* and *Positive predicted value,* was also evidenced in Random Forest, Logistic Regression, Decision Tree, Naive Bayes, Support Vector Machines and Neural Networks. This suggests that the new predictive variables proposed in this work allowed us to improve the OOS detection. To determine the information gain contribution of the new predictor variables, we analyzed their importance. [7] demonstrated the importance of the sales predictor variables in the detection of OOS. [10] also showed the importance of sales records to detect the occurrence of OOS using POS data through the recognition of sales patterns. Recently, [11] presented their HMM to detect OOS occurrence, which was calibrated with sales records from POS data. Our work showed that the top 15 importance variables include sales, inventory, and ordering features (see Figure 7 and Table 5). Of these 15 predictor variables, 11 variables corresponded to new predictor variables proposed in this work (73%): five variables corresponded to inventory features (33%), three variables corresponded to ordering features (20%), and three variables corresponded to sales features (20%). Finally, for further analysis, the classification algorithms that we chose were the best single classification algorithm (Random Forest) and the best global classification algorithm (Ensemble).

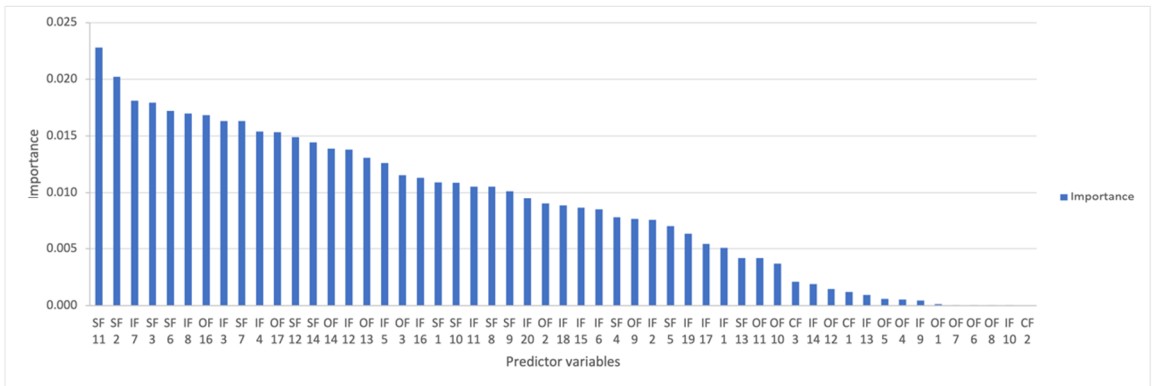

**Figure 7.** Predictor variables importance ranking.

**Table 5.** Top 15 Predictor variables ranking.

| Predictor Variables | Attribute Description | Importance | Rank | References |
|---|---|---|---|---|
| SF 11 | The number of days that a product has not sold any single unit in a period/Total number of days in a period (percent) | 0.02280 | 1 | New attribute |
| SF 2 | Average number of items sold in a period | 0.02022 | 2 | [7,8] |
| IF 7 | Average inventory level using only the days that a product has positive stock in a period | 0.01811 | 3 | New attribute |
| SF 3 | Standard deviation of items sold in a period | 0.01793 | 4 | [7,8] |
| SF 6 | Average number of sales using only the days that a product made a sale in a period | 0.01721 | 5 | [7,8] |
| IF 8 | Standard deviation inventory level using only the days that a product has positive stock in a period | 0.01700 | 6 | New attribute |

**Table 5.** *Cont.*

| Predictor Variables | Attribute Description | Importance | Rank | References |
|---|---|---|---|---|
| OF 16 | PO reception qty (deliver): number of items to deliver on a period to the D.C. for a specific store | 0.01685 | 7 | New attribute |
| IF 3 | Average inventory level in a period | 0.01634 | 8 | New attribute |
| SF 7 | Standard deviation of sales using only the days that a product made a sale in a period | 0.01632 | 9 | [7,8] |
| IF 4 | Standard deviation inventory level in a period | 0.01540 | 10 | New attribute |
| OF 17 | PO reception qty (received): number of items received on a period to the D.C. for a specific store | 0.01534 | 11 | New attribute |
| SF 12 | The number of days that a product hasn't sold any single unit in a period | 0.01491 | 12 | New attribute |
| SF 14 | Items sold for the specific day/Average number of items sold in a period (percent) | 0.01445 | 13 | New attribute |
| OF 14 | The number of days that a product has been ordered in a period/Total number of days in a period (percent) | 0.01389 | 14 | New attribute |
| IF 12 | Standard deviation Stock adjustment in a period | 0.01381 | 15 | New attribute |

### 4.4. Imbalance Data Problem

To implement the machine learning based system in the manufacturer, it was necessary to explore the effect of using imbalanced real-world data in the predictive performance of the classification algorithm. As shown in Tables 6 and 7, we balanced the DS.CS_2 data by over-sampling the minority examples (OOS) and under-sampling the majority examples (EXIST), to then run both classification algorithm previously selected. For both cases, we tested different balancing percentages. In the over-sampling strategy, we increased the number of samples of the OOS class until they represented 50% of the data. In the under-sampling strategy, we decreased the number of samples of the EXIST class until they represented 50% of the data.

**Table 6.** Data balancing evaluation: Random Forest.

| Dataset | Statistics | Random Forest | | | | | |
|---|---|---|---|---|---|---|---|
| | | Accuracy | Sensitivity | Pos. Pred. Value | Specificity | Neg. Pred. Value | F-Measure |
| Not Balanced Data | mean | 0.93430 | 0.46316 | 0.85453 | 0.99058 | 0.93924 | 0.60073 |
| DS.CS_2 | sd | 0.00850 | 0.05119 | 0.05436 | 0.00383 | 0.00912 | 0.05273 |
| Balanced Data | mean | 0.93457 | 0.50002 | 0.81394 | 0.98628 | 0.94317 | 0.61948 |
| Oversampling 20% | sd | 0.00827 | 0.05927 | 0.06509 | 0.00538 | 0.00853 | 0.06204 |
| Balanced Data | mean | 0.93608 | 0.56350 | 0.77449 | 0.98045 | 0.94973 | 0.65236 |
| Oversampling 30% | sd | 0.00831 | 0.06287 | 0.05794 | 0.00560 | 0.00869 | 0.06030 |
| **Balanced Data** | **mean** | **0.93650** | **0.58342** | **0.76570** | **0.97855** | **0.95182** | **0.66225** |
| **Oversampling 40%** | **sd** | **0.00804** | **0.05603** | **0.06268** | **0.00669** | **0.00795** | **0.05917** |
| Balanced Data | mean | 0.93530 | 0.58124 | 0.75573 | 0.97750 | 0.95150 | 0.65710 |
| Oversampling 50% | sd | 0.00853 | 0.06036 | 0.06206 | 0.00665 | 0.00862 | 0.06120 |
| Balanced Data | mean | 0.92662 | 0.57491 | 0.68717 | 0.96841 | 0.95049 | 0.62605 |
| Oversampling 20% | sd | 0.00926 | 0.06244 | 0.06812 | 0.00911 | 0.00808 | 0.06515 |
| Balanced Data | mean | 0.90437 | 0.66823 | 0.54283 | 0.93247 | 0.95948 | 0.59904 |
| Oversampling 30% | sd | 0.01205 | 0.06372 | 0.05755 | 0.01327 | 0.00842 | 0.06048 |
| Balanced Data | mean | 0.86063 | 0.75208 | 0.41646 | 0.87361 | 0.96742 | 0.53607 |
| Oversampling 40% | sd | 0.01772 | 0.06137 | 0.04482 | 0.02103 | 0.00842 | 0.05181 |
| Balanced Data | mean | 0.78274 | 0.82056 | 0.30694 | 0.77814 | 0.97351 | 0.44676 |
| Oversampling 50% | sd | 0.02543 | 0.05332 | 0.03150 | 0.03011 | 0.00741 | 0.03960 |

**Table 7.** Data balancing evaluation: Ensemble.

| Dataset | Statistics | Ensemble | | | | | |
|---|---|---|---|---|---|---|---|
| | | Accuracy | Sensitivity | Pos. Pred. Value | Specificity | Neg. Pred. Value | F-Measure |
| **Not Balanced Data** | **mean** | **0.93967** | **0.56735** | **0.80480** | **0.98358** | **0.95072** | **0.66553** |
| **DS.CS_2** | **sd** | **0.00880** | **0.05451** | **0.05792** | **0.00590** | **0.00901** | **0.05616** |
| Balanced Data | mean | 0.93123 | 0.44123 | 0.82372 | 0.98863 | 0.93801 | 0.57465 |
| Oversampling 20% | sd | 0.00971 | 0.06576 | 0.06237 | 0.00510 | 0.01100 | 0.06402 |
| Balanced Data | mean | 0.92754 | 0.39072 | 0.82858 | 0.99031 | 0.71529 | 0.53103 |
| Oversampling 30% | sd | 0.01015 | 0.07590 | 0.06494 | 0.00465 | 0.40148 | 0.06999 |
| Balanced Data | mean | 0.91935 | 0.26699 | 0.87367 | 0.99547 | 0.92093 | 0.40899 |
| Oversampling 40% | sd | 0.01105 | 0.05479 | 0.07258 | 0.00269 | 0.01076 | 0.06244 |
| Balanced Data | mean | 0.90772 | 0.12876 | 0.90673 | 0.99864 | 0.90767 | 0.22550 |
| Oversampling 50% | sd | 0.01291 | 0.06572 | 0.13297 | 0.00197 | 0.01254 | 0.08796 |
| Balanced Data | mean | 0.92604 | 0.61306 | 0.66034 | 0.96262 | 0.95522 | 0.63582 |
| Oversampling 20% | sd | 0.00957 | 0.06414 | 0.05996 | 0.01010 | 0.00896 | 0.06198 |
| Balanced Data | mean | 0.89986 | 0.70397 | 0.51847 | 0.92288 | 0.96383 | 0.59714 |
| Oversampling 30% | sd | 0.01580 | 0.05737 | 0.06056 | 0.01619 | 0.00864 | 0.05892 |
| Balanced Data | mean | 0.85877 | 0.76567 | 0.40731 | 0.86970 | 0.96946 | 0.53174 |
| Oversampling 40% | sd | 0.01623 | 0.04774 | 0.03135 | 0.01833 | 0.00753 | 0.03784 |
| Balanced Data | mean | 0.81420 | 0.81740 | 0.34075 | 0.81412 | 0.97430 | 0.48099 |
| Oversampling 50% | sd | 0.02507 | 0.04362 | 0.03980 | 0.02811 | 0.00760 | 0.04162 |

For our data, we observed that the Ensemble approach was able to effectively handle the unbalance problem, observing that no improvements were obtained when incorporating balancing techniques (Table 7). However, at the individual classifier level (Random Forest), the balancing approach had a positive effect.

In the over-sampling strategy, by training the Random Forest classifier with a greater number of examples of the minority class variable OOS, the *Sensitivity* metric improves since the classifier can detect a greater number of OOS events. This metric improves directly as the percentage of OOS examples in the dataset increases. Consequently, because the participation of the majority class variable EXIST decreases in the dataset, the metric *Positive predicted value* worsens. These results are consistent with the tradeoff challenge between both metrics. For the under-sampling strategy, the same previous behavior is observed. The participation of the minority class variable examples (OOS) increases in the dataset since the majority class variable's examples decrease. The differences observed between both strategies underlie the magnitude of the *Sensitivity* improvement versus the worsening of *Positive predicted value*. Our goal is to achieve the best tradeoff between both performance metrics. The prediction performance obtained from training the Random Forest with the balanced DS.CS_2 over-sampling (40%) dataset, presented the best tradeoff between *Sensitivity* and *Positive predicted value*, with the highest *F-measure* value 0.66225. This result was similar to that obtained by the Ensemble without balanced data (highest *F-measure*, 0.66553). Consequently, for the implementation of the prediction model in the real-world setting, we chose to test the Ensemble algorithm without data balance and the Random Forest algorithm, with balanced data.

*4.5. Machine Learning Based System implementation: Real-World Setting*

Based on the previous results, we developed and implemented two systems based on the classification algorithms to automatically detect products missing from the shelf. The first predictive system applied an Ensemble algorithm and, the second applied a Random Forest classification algorithm. The Ensemble was trained with unbalanced data, and the Random Forest was trained with a 60% EXIST and 40% OOS distribution dataset through oversampling the minority class. The Ensemble training dataset was composed of 2296 records, 53 predictor variables, and a binary class variable obtained through store physical audits. The Random Forest training dataset was composed of 4941 records, 53 predictor variables, and a binary class variable obtained through store physical audits. To implement this system in a real-world scenario, we proposed a framework consisting of six stages (see Table 8).

**Table 8.** Machine Learning Based System framework.

| Stages: | Stage 1 | Stage 2 | Stage 3 | Stage 4 | Stage 5 | Stage 6 |
|---|---|---|---|---|---|---|
| Activity: | Daily POS data extraction and consolidation from the manufacturer´s IS | Data processing | Computation of Predictor variables | OOS prediction | Preparation of daily reports for the manufacturer´s store visitors' staff | Store visits to correct potential OOS events |
| Resource: | IT staff | IT staff | IT staff | IT staff | IT staff | Store visitors' staff |

Once our OOS prediction system was implemented in the manufacturer, it was validated with new real-life data. The retail department used this system to deliver information on possible OOS events to its store visitor's staff. Seven stores were chosen to implement the proposed framework. At the beginning of the day, the IT area prepared an OOS prediction report delivered to the store visitor's staff who visited the stores daily, reviewing the information provided, indicating the actual status (OOS/EXIST) of each product in their report. This physical store audit was carried out over four consecutive weeks. We collected this information in a new dataset (Pa.D_Val.) which contained 1764 records, with seven attributes (date, store, product, sale, stock, predictions, and physical store audit). The result of this new physical audit is presented in Table 9.

**Table 9.** Physical store audit: Machine Learning Based System real-world validation.

| Dataset | Description | Observed Class | | Total | Rate | Category | Product Characteristics |
|---|---|---|---|---|---|---|---|
| | | EXIST | OOS | EXIST + OOS | OOS | | |
| Pa.D_Val | Store physical audit dataset: Machine Learning system validation | 1713 | 51 | 1764 | 2.9% | Fresh fruits and vegetables | Non-perishable food (Nuts and Dried fruit) |

In this new physical store audit, we found a lower rate of OOS events (2.9%). This result presents an additional challenge to our prediction system to detect this low occurrence of OOS events. The performance metrics that we evaluated to determine the prediction performance of our system in a real-life scenario were *Sensitivity* and *Positive predicted value*. The performance results of the machine learning based system are presented in Table 10.

**Table 10.** Machine Learning Based System Real-world Performance Metrics.

| DETECTION SYSTEM | Performance Metrics | | |
|---|---|---|---|
| | Sensitivity | Pos. Pred. Value | F-Measure |
| Machine Learning based system: Real-world OOS prediction (Ensemble) | 0.58000 | 0.77333 | 0.66286 |
| Machine Learning based system: Real-world OOS prediction (Random Forest) | 0.68000 | 0.72340 | 0.70103 |

The results for the performance metrics obtained in the real-world validation (Table 10) are consistent with the results presented in Tables 6 and 7. The *Sensitivity* and *Positive predicted value* for the real-world validation of the Machine Learning based system Ensemble were 0.58000 and 0.77333, and the Random Forest performance metrics were 0.68000 and

0.72340, respectively. The Ensemble *F-measure* value was 0.66286 and Random Forest was 0.70103. At the implementation of these classifiers in a real-world setting, we observed that the approach using Random Forest with balanced data presented a better performance compared to Ensemble with unbalanced data. In the previous Random Forest balance data evaluation, for the oversampling (40%), the *Sensitivity* and *Positive predicted value* were 0.58342 and 0.76570, respectively, and its *F-measure* value was 0.66225. These results validate that the proposed Machine Learning based system could be implemented in a manufacturing company, contributing to the automatic detection of OOS events in the retail industry.

## 5. Discussion

The use of sales features to predict the occurrence of OOS has been reported [7,10,11]. If a product is not available on the shelf, it could cause sales loss compared to a normal product. In this work, novel inventory features variables were proposed to detect the occurrence of OOS. One of the reasons for incorporating these variables is for the existence of phantom inventories [10,29,46]. This phenomenon could cause that the automatic purchasing system (CAO) does not issue purchase orders because of the existence of a "theoretical" zero demand with a "positive" inventory in the information system. This scenario led to an OOS situation with no replacement purchases orders until the problem is detected and corrected. Accordingly, the issuance of purchase orders is also important. We proposed to incorporate novel ordering features variables. We suggested that a distortion in the behavior of issuing purchase orders, such as frequency or quantity, could be an OOS predictor.

In the context of inventory features, there is information inaccuracy in inventory systems [29]. As a result, inventory records appear to overestimate the number of units present in the store, rendering inventory data untrustworthy. However, we proposed to include certain inventory features as OOS predictors variables to take advantage of this situation. We suggest first if there is low or zero inventory, there is a high probability that there is no product in the store (because the inventory system records tend to overestimate the actual product quantity). Second, phantom stockouts [46] and information inaccuracy in inventory systems [29] are real problems. If there are any records in the information systems related to their detection/correction, these records could be predictors of OOS. In our case study, inventory adjustments records could be found, for example, once a difference between physical inventory and inventory records is detected, this difference is corrected through an adjustment. This adjustment could be positive (inventory records quantity are less than physical inventory) or negative (inventory records quantity are greater than physical inventory). Based on these arguments, we proposed that product adjustments are recorded as novel predictors' variables.

The EDA allowed us to reinforce our theoretical analysis to propose the incorporation of new predictor variables. As we observed in Section 4.2, we showed a positive correlation between new predictor variables proposed in this work and the average sales variable, which has been used in previous works [7,8]. The PCA result showed the problem's complexity. Not identifying and grouping the class variable suggests the need to use techniques such as machine learning classification algorithms to address OOS detection.

In Section 4.3, we compared the performance of two data sets, DS.CS_1 and DS.CS_2 in six different single classification algorithms and an Ensemble. DS.CS_1 contains some of the top variables presented in [7] and the DS.CS_2 besides includes the new predictor variables proposed in this work. Random Forest presented the best performance as for trade-off between *Sensitivity* and *Positive predicted value* for both datasets (highest value for *F-measure* metric), compared to single classifiers. The Ensemble showed the best overall performance. These results were consistent with those presented in [7]. To determine the contribution of the predictor variables in the algorithm performance, we evaluated the importance of the variables. As observed in Table 5, 73% of the top 15 predictor variables correspond to novel predictors proposed in this work, belonging to inventory, ordering,

and sale features. Furthermore, DS.CS_2 presented a better performance for both *Sensitivity* and *Positive predicted value* than DS.CS_1. The DS.CS_2 *F-measure* value was 70% higher than DS.CS_1. This suggests that the new predictor variables proposed in this work are relevant for the detection of OOS and contribute to improving the two most significant performance metrics in this case study. One of the main challenges of detecting OOS in retail is the complexity and dynamism of this industry. There are multiple roots causes for a product missing on the shelf. Therefore, our proposal to expand the problem scope, including the manufacturer, retailer DC and retailer store, allowed us to improve the OOS detection and enhance the classification algorithms' performance.

In a real-world scenario, data imbalance is a challenge, where the class to be predicted is less represented in the dataset. For machine learning classification algorithms, this could affect their performance since there are fewer examples of the class of interest in the training data set. In this case study, the percentage of OOS determined through physical audits was 10%. This result was consistent with those presented in previous papers [7,58]. This means that the class to be predicted represents only 10% of the total records in the data set. To overcome this challenge, we proposed using data balancing strategies, such as oversampling (increasing the number of examples from the minority class) or undersampling (reducing the number of examples from the majority class. Considering that one of our goals was to improve the tradeoff between performance metrics *Sensitivity* and *Positive predicted value*, given the results presented in 4.4, we proposed using the oversampling technique to increase the OOS sample size from 10% to 40% for training a single Random Forest, and an Ensemble without balanced data. With the data balancing strategy, Random Forest obtained its best tradeoff between *Sensitivity* and *Positive predicted value* (*F-measure* value = 0.66225), having the advantage of not losing critical information in the process.

Finally, in 4.5, we presented the implementation of this model in a manufacturer company and the validation in a real-world scenario. Table 8 presented the proposed framework implemented in this case study. This framework was successfully implemented in the manufacturing company, managed by their IT staff, who prepared daily reports delivered in the morning to the sales and the store visit staff. The proposed model validation was carried out for four consecutive weeks. The store visit team validated the OOS detected by the machine learning-based system. We obtained an important tradeoff between *Sensitivity* and *Positive predicted value*. For Random Forest, *Sensitivity* value was 0.68000, *Positive predicted value* was 0.72340 and *F-measure* was 0.70103. For the Ensemble, *Sensitivity* value was 0.58000, *Positive predicted value* was 0.77333 and *F-measure* was 0.66286. It is important to highlight that in the real-world model validation, Random Forest obtained a 5.8% higher *F-measure* value than what we obtained in the data balancing evaluation for an oversampling of 40%. This result is even more relevant because the presence of OOS during the validation study was 2.9%, which supports the importance of applying data balancing strategies in this case study. However, the Ensemble presented slightly lower performance in real world application.

When comparing our results with those obtained in previous works [7], presented an average of approximately 80% in *Accuracy* and 22% *Support*. It is important to mention that the author indicated that this last metric was an estimate. In [7] *accuracy* metric corresponded to our *Positive predicted value* metric, and *Support* metric corresponded to our *Sensitivity* metric. Although in [7], the OOS detection is performed directly with the retailer, having access to its transactional data information (point of sale data), considering for example, different categories or promotional products and market share of the product in the category to which it belongs. We observe that, in our work, the definition and inclusion of new predictor variables have contributed to obtaining a better tradeoff between the performance metrics of interest. In [11] they also used POS historical data obtained directly from the retailer and a Hidden Markov Model to detect OOS. The results reported in that work were *Power of detection* (63.48%) and *False alarms* (15.52%). The metric *Power of detection* corresponded to our *Sensitivity* metric, and (1 − *False alarms*) corresponded to

our *Positive predicted value* metric. We observed that the performance metrics reported in both works were similar. In [11], the proposed model was implemented in the retailer, being able to obtain a significant volume of historical point of sale data. Dissimilar, our proposal addresses the manufacturer's perspective. One of the most important challenges to overcome is that the manufacturer can only access a limited fraction of the retailer´s historical data. Therefore, the high detection power and low rate of false alarms obtained in our work are fundamentally due to the new predictive variables proposed through domain knowledge.

## 6. Conclusions

The OOS detection in the grocery retail sector has focused on what we categorize as the retailer perspective. In this work, we addressed this problem from a manufacturer's perspective, contributing to advance in collaborative retailer–manufacturer information integration.

It is possible to use POS data obtained from the manufacturer's information system, which is shared by the retailer, to detect the occurrence of OOS. It is also feasible to implement a machine learning based system in manufacturers, which allows them to prepare management reports for their store visitor's staff, collaborating with the retailer's staff to face the OOS problem.

Due to the complexity of this problem, it is important to incorporate new variables that allow us to make better predictions. For this, it is essential to know the case study, being a field expert. To improve the classifier performance metrics, it is also necessary to consider the data imbalance, a typical problem in data obtained from real life. Increasing the scope of the problem, proposing new predictor variables and data balancing strategies or classifier ensembles, allowed us to develop a machine learning model that obtains an important percentage of precision (72%) and a significant amount of OOS (68%) in a real-world scenario (a Random Forest with balanced data). These results have an important impact on the retail operation. Prior to the implementation of this tool, to detect OOS, the manufacturer must allocate significant staff resources that audit the stores through a route plan, defined by the store size, location, and sales volume. Although OOS are a significant problem, these events represent 10% on average. Therefore, for the manufacturer, it is to be expected that a high number of visits do not detect OOS in a timely manner. After implementing our classification algorithm, the results obtained changed the way that the manufacturer manages its store visiting staff. It established visits to stores prioritizing the alerts reported by the classification algorithm, allowing it to increase the efficiency of its staff. According to the information provided by the manufacturer, after incorporating this tool into its operational management, the store visit staff double the scope of its store coverage (stores visited per week), being able to improve the on-shelf availability (+34%).

The next step for further research is implementing this model in other product categories, such as groceries, dairy, and others. Moreover, another interesting prospect of investigation is testing the model predictive performance considering a new role for the store audit team. In addition to reporting the on-shelf availability, they should correct the OOS events and potential errors in the information system. We also propose future research related to increasing the scope of the problem, incorporating new variables and operational processes. In this work, we have shown that new inventory predictor variables are relevant. A clear increase in scope is to incorporate differentiated inventory data between the shop floor and the backroom. This could contribute to improving the model predictive performance and identifying potential OOS root causes. Finally, it is important that, in real life scenarios, the models proposed could be implemented by the end user; in this case, the manufacturer. For this reason, the implementation of a single classifier, such as Random Forest, was easier to implement in the workflow of the manufacturer's company. However, in the first part of our study, the Ensemble obtained the best performance; therefore, we propose as future research to perform studies with other strategies of

Ensemble, to continue looking for performance improvements and overcome the challenge of unbalance real world data.

**Author Contributions:** Conceptualization, J.M.R.A., G.A.R. and M.G.; methodology, J.M.R.A., G.A.R. and M.G.; software, J.M.R.A.; validation, J.M.R.A., G.A.R. and M.G.; formal analysis, J.M.R.A., G.A.R. and M.G.; investigation, J.M.R.A., G.A.R. and M.G.; resources, J.M.R.A. and G.A.R.; data curation, J.M.R.A.; writing—original draft preparation, J.M.R.A.; writing—review and editing, J.M.R.A., G.A.R. and M.G.; visualization, J.M.R.A.; supervision, G.A.R. and M.G.; project administration, J.M.R.A.; funding acquisition, J.M.R.A. and G.A.R. All authors have read and agreed to the published version of the manuscript.

**Funding:** The authors were supported by CONICYT-Chile under grant CONICYT Doctoral scholarship (2015-21151606) (J.M.R.A.), ANID FONDECYT grant number 1180706 (G.A.R), ANID PIA/BASAL FB0002 (G.A.R.), and Centro de Modelamiento Matemático (CMM), FB210005, BASAL funds for centers of excellence from ANID-Chile (M.G.).

**Data Availability Statement:** The data presented in this study are available on request from the corresponding author.

**Conflicts of Interest:** The authors declare no conflict of interest.

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
