# Peer review of "Predicting Out-of-Stock Using Machine Learning: An Application in a Retail Packaged Foods Manufacturing Company"

_electronics, doi:10.3390/electronics10222787_

Round 1

Reviewer 1 Report

  • The revised version of the paper is well written, analysed and discussed.
  • However, please check some English.

Reviewer 2 Report

The paper sounds globally good, I simply suggest you to provide more information in the experimental part by answering the following questions :

Which hyper-parameters are tuned in which range of values and do you do it ?
How many folds do you use during the tuning process ?

Reviewer 3 Report

This paper is good in some aspects, however, the issue is the novelty of the proposed method which is simple 

I would suggest adding the RNN(STEM) to your classifiers then performing fusion on the results of all classifiers. This way will give more robustness. 

This paper will help you in terms of fusion 

Ensemble based systems in decision making

https://ieeexplore.ieee.org/document/1688199

Round 2

Reviewer 3 Report

The authors have successfully addressed the comments